# Flocking in complex environments—Attention trade-offs in collective information processing

**Parisa Rahmani**[1,2], **Fernando Peruani**[3], **Pawel Romanczuk**[2,4]*

**1** Department of Physics, Institute for Advanced Studies in Basic Sciences (IASBS), Zanjan, Iran, **2** Institute for Theoretical Biology, Department of Biology, Humboldt Universität zu Berlin, Berlin, Germany, **3** Université Côte d'Azur, Laboratoire J.A. Dieudonné, UMR 7351 CNRS, Parc Valrose, F-06108 Nice Cedex 02, France, **4** Bernstein Center for Computational Neuroscience Berlin, Berlin, Germany

* pawel.romanczuk@hu-berlin.de

**Data Availability Statement:** All figure data files are available from the figshare database (accession number https://doi.org/10.6084/m9.figshare.11889024.v1).

## Abstract

The ability of biological and artificial collectives to outperform solitary individuals in a wide variety of tasks depends crucially on the efficient processing of social and environmental information at the level of the collective. Here, we model collective behavior in complex environments with many potentially distracting cues. Counter-intuitively, large-scale coordination in such environments can be maximized by strongly limiting the cognitive capacity of individuals, where due to self-organized dynamics the collective self-isolates from disrupting information. We observe a fundamental trade-off between coordination and collective responsiveness to environmental cues. Our results offer important insights into possible evolutionary trade-offs in collective behavior in biology and suggests novel principles for design of artificial swarms exploiting attentional bottlenecks.

## Author summary

Understanding how consensus is reached and information is processed within a collective is fundamental to many aspects of social dynamics in animals and humans. It is widely accepted that high connectivity among individuals facilitates group consensus, and being in a group provides benefits to individuals through social information about the environment provided by other group members. We show that this does not hold for collectives in complex environments: Limited attention capacity, that severely reduces connectivity among individuals, is highly beneficial for global coordination. However, this comes at a price: Collectives outperform isolated individuals in responding to the environment only at sufficiently high attention capacities, where global coordination breaks down. Thus, we demonstrate a fundamental trade-off in collective behavior between social coordination and responsiveness to environmental cues. Our work demonstrates the importance of sensory and cognitive limitations for the emergence and function of animal collectives, and poses fundamental questions about co-evolution of social behavior and individual attention capacity. The observed trade-off in collective information processing has implications for human social systems and for the design of robotic swarms operating in complex environments.

**Funding:** P. Romanczuk acknowledges funding by the Deutsche Forschungsgemeinschaft (DFG, German Research Foundation) through RO 4766/2-1. P. Romanczuk acknowledges funding by the Deutsche Forschungsgemeinschaft (DFG, German Research Foundation) under Germany's Excellence Strategy – EXC 2002/1 "Science of Intelligence" – project number 390523135 P. Rahmani was supported by German Academic Exchange Service (DAAD) and by the Ministry of Science, Research and Technology of Iran. F. Peruani was supported by the Agence Nationale de la Recherche via project BactPhys, Grant No. ANR-15-CE30-0002-01. The funders had no role in study design, data collection and analysis, decision to publish, or preparation of the manuscript.

**Competing interests:** The authors have declared that no competing interests exist.

## Introduction

Consensus formation, coordination and collective response to environmental cues are important aspects in collective behavior of many interacting agents in biology, physics, robotics and computational social science. The understanding of these processes is fundamental for a better comprehension of collective intelligence that confers groups the ability to solve problems collectively using strategies which are beyond the reach of single individuals [1]. Crucial for understanding benefits of collective behavior, is the understanding of the mechanisms underlying collective information processing: How new information is acquired, how information is shared and combined within the collective, and how the collective deals with conflicting information are among the most compelling and elusive questions on the self-organization of collectives. Generic flocking models (see e.g. [2–5]) allow to study the interplay between emergent collective behaviors and collective information processing in a dynamical system setting. For example, using flocking models it has been demonstrated that only a small fraction of informed individuals is sufficient to accurately guide large collectives [6–8]. It was also shown that groups are able to collectively track dynamic environmental gradients not detectable by individuals [4, 9], or that groups can make efficient consensus decisions in conflict situations without any implicit knowledge about the majority-minority relationships [6, 10]. Only recently, predictions of such models on fundamental decision bifurcations in spatial movement decisions have been confirmed in the collective migration of baboon groups [11].

A fundamental aspect of the self-organization of collectives—from collective decision-making to consensus formation, including coordinated movements—is that individuals are limited in terms of perception and cognition. Without direct access to the state of the whole group they must rely on local information [12–15]. The emergent collective patterns in agent-based models have been shown to depend strongly on the field of view of individuals [16, 17], and more generally on what local information individuals pay attention to [18, 19]. Furthermore, even for a strongly limited field of view, the sensory input of individual agents may contain a large number of social and non-social cues. However, social interactions in animal groups appear to be restricted to a rather low number of neighbors [20, 21]. On the one hand, this suggests additional cognitive constraints on the processing of available sensory information, which is also in-line with a wide range of experimental results on limited capacity for visual tracking of multiple objects in animals and humans [22–26]. On the other hand, it has been shown in generic flocking models that the emergent, large scale collective behavior depends strongly on how many neighbors a given individual can pay attention to [27–29]. However, to our knowledge the explicit role of cognitive constraints on collective information processing has not been systematically explored.

Up-to-date most theoretical and empirical research focused on information sharing and collective decision making, in idealized, laboratory-like environments, including the works mentioned above, discussing cognitive and sensorial limitations (see e.g. [6, 10, 30, 31]). However, collectives in realistic scenarios need to cope with complex environments with a large number of potentially informative or distracting environmental cues (see Fig 1a and [32]). Whereas recently it was shown using minimal flocking models that self-organized, collective behaviors are strongly affected by complex environments [33, 34], it remains open how complex environments impact collective information processing, which is the main question we focus on here. More specifically, we investigate the emergence of collective behaviors in a generic model of socially interacting agents in a complex environment containing many potentially dangerous sites. Individuals try to avoid these sites, while at the same time trying to coordinate with their neighbors. In addition, some informed individuals have also private information on a global preferred direction of migration, which may be in conflict with the

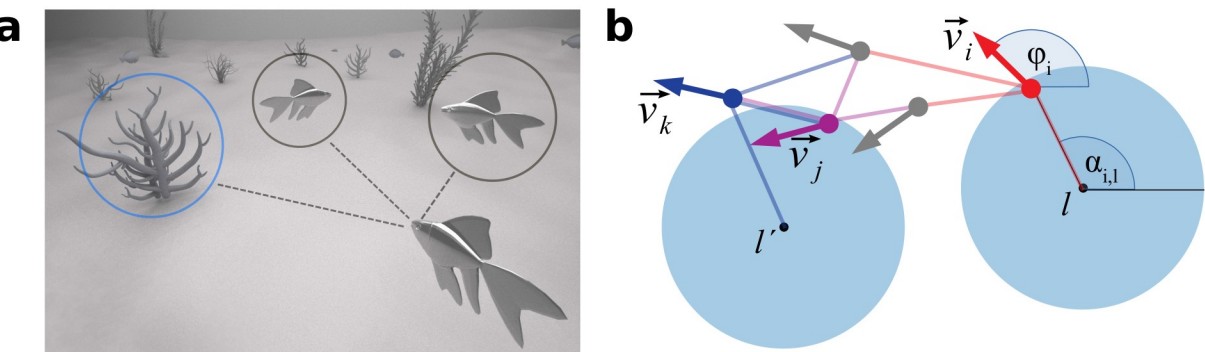

**Fig 1. Attention trade-off in collective behavior. a**: Schematic visualization of attention trade-off in collective behavior in complex environments. The focal individual can only pay attention to $k = 3$ nearest objects—other agents or non-social environmental features—simultaneously. **b**: Visualization of different situations that may occur in the model. The arrows indicate the velocity vectors $\vec{v}$ of the different agents. The small black circles indicate the location of danger sites $l$ and $l'$ with their repulsion zones shown in blue. Agent $i$ (red) reacts to the danger site (DS) $l$ as two conditions are met simultaneously: DS $l$ is in $i$'s kNO, and agent $i$ is also within the corresponding repulsion zone. Agent $j$ (magenta) does not react to DS $l'$ since it's attention slot is already filled with three other agents (one blue and two gray). Agent $k$ (blue) perceives DS $l'$ but does not react to it, because it is outside of the repulsion zone. It only reacts to two other neighbors (gray and magenta) and aligns with them.

local environmental cues. Our main aim is to explore how the cognitive, and/or sensory constraints of the individuals affect group-level coordination, information exchange, and collective response to environmental cues. We note that in the following we will refer to *complex* environments, containing high density of distraction sites, also as *heterogenous* environments, in order to distinguish them from *homogeneous* environments without any distraction sites.

Our results show that in heterogeneous environments, strongly limited attention capability of individual agents results in higher accuracy with respect to large-scale coordination, which is in stark contrast to previous results obtained in simple environments [28, 29]. This is caused by a dynamical, spatial "echo chamber"-like effect, where individual attention becomes saturated by social information and non-social cues are largely ignored. However, if these non-social cues provide important information about potential environmental dangers, the emergent dynamical "echo chambers" become strongly detrimental to the ability of the collective to safely navigate the environment. Note that information exchange through social interactions is typically believed to be beneficial for the collective [9, 35]. Here, our analysis shows that below a critical threshold in attention capacity, groups perform worse at acquiring new information about the environment than non-interacting agents. This is due to the emergent self-isolation from environmental cues, which is exactly what facilitates group coordination in complex environments. Our findings not only suggest a fundamental trade-off in collective behavior in natural systems, but also provide important insights for the design of communication in artificial, distributed systems, such as robotic swarms.

## Results

### Model

We consider a flocking model consisting of $N$ agents moving in a two-dimensional environment of size $L \times L$ with periodic boundary conditions. In addition, we assume the presence of $N_{\text{inf}}$ informed individuals with private information about a preferred direction of motion $\hat{u}_p$. The informed fraction of the collective is $R_{\text{inf}} = N_{\text{inf}}/N$.

The environment contains non-social cues, which represent features of the environment that solitary agents in general try to avoid, as they, for example, signal potential dangers. We

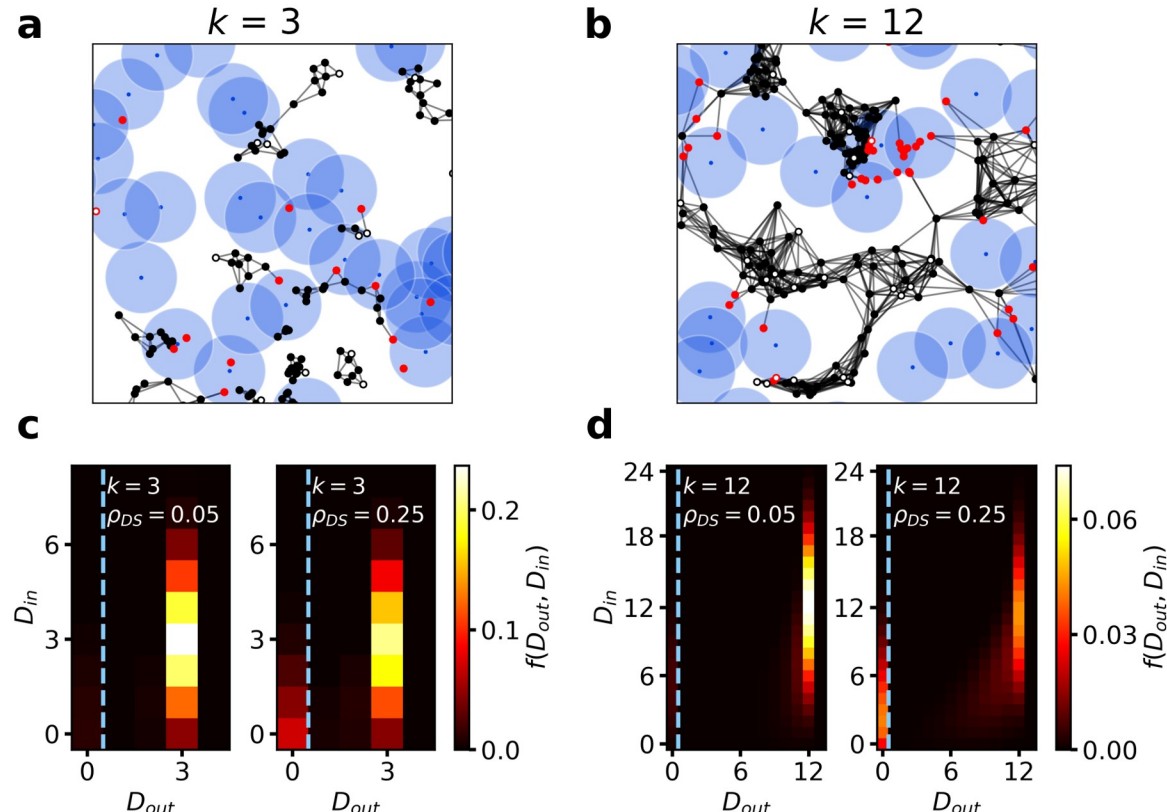

**Fig 2. Emergent interaction networks. a,b**: Examples of social interaction networks for $k = 3$ (**a**) and $k = 12$ (**b**) at $\rho_{DS} = 0.25$. The black symbols indicate socially interacting agents, whereas the red symbols indicate agents responding to a DS. The lines indicate the (non-directed) interaction network. Filled circles represent uninformed agents, empty circles indicate agents informed about the preferred direction of migration. The DS positions shown by blue dots, are surrounded by a disc-like repulsion zones (light blue). For clarity, only a portion of the respective simulation box is represented here, see S1 Fig. for the full snapshots. **c,d**: In-out degree distributions for the emergent social interaction networks for low attention limit $k = 3$ (**c**) and high attention limit $k = 12$ (**d**) at low and high DS densities ($\rho_{DS}$ = 0.05, and $\rho_{DS}$ = 0.25). The vertical dashed lines are for visual guidance to distinguish the subpopulations with $D_{out} = 0$ corresponding to agents responding to DSs (left of the vertical line). At high density of DSs this distribution is clearly bimodal with two peaks at $D_{out} = 0$ and $D_{out} = k$. By increasing DS density number of agents with $D_{out} = 0$ increases. These agents have a lower in-degree compared to non-responders, which contributes to the self-isolation of the collective from environmental cues at low $k$ values. For all panels: $R_{inf} = 0.1$.

will refer to these disrupting cues as "danger sites" (or distraction sites) in short as DS. Each DS is surrounded by an effective repulsion zone of radius $r = 1$. The environment contains $N_{DS}$ randomly distributed DSs at fixed positions. In particular at high DS densities $\rho_{DS} = N_{DS}/L^2$, the corresponding repulsion zones may overlap (see Fig 2a and 2b). We note that the agents and the danger sites (DSs) are assumed to be point-like, as in many agent-based models for collective movement, e.g. Vicsek-type models [3]. This corresponds to the scenario where the sensory ranges are large compared to the physical size of moving agents and DSs.

Based on experimental observations, it was suggested that animals interact with a limited number of conspecifics [14, 20, 21]. Motivated by these findings, and with intention of explicitly studying the impact of limited attention in a generic flocking model, we assume that each agent can pay attention only to $k$ nearest objects ($kNO$) in its vicinity, irrespective whether it is another agent (social cue) or a DS (non-social cue). Thus, $k$ can be interpreted as the number of available attention slots for each agent. The parameter $k$ quantifies the individual attention capacity.

Each agent moves with a constant speed $v_0$ and reacts to neighbors and DSs through corresponding changes in its direction of motion $\hat{u}_i = (\cos \varphi_i, \sin \varphi_i)^T$ defined by a polar orientation angle $\varphi_i$. All agents—informed and uninformed—turn away from DSs when two conditions are met simultaneously: 1) the agent perceives the DS—that means that the DS in question, say $l$, is within its $k$ nearest objects—and 2) the distance between the agent and DSs ($d_{il} = |\vec{x}_i - \vec{x}_l|$) is smaller than the radius of its repulsion zone. In all other cases, individuals ignore the DSs and coordinate their motion with other individuals within their kNO by aligning their direction of motion with the average direction of their neighbors. Informed individuals exhibit an additional bias to orient towards the preferred direction of motion that we denote $\hat{u}_p$. Throughout this work, the preferred direction of motion of informed individuals will be along the $x$-axis: $\hat{u}_p = (1, 0)^T$. We emphasize that the response to DSs, once detected, dominates all other behavioral responses of individuals, whether informed or uninformed. The specific formulation of the mathematical model in terms of stochastic differential equations, together with the parameters used, is given in Methods.

The finite attention capacity to $k$ nearest objects leads to a natural competition between social and non-social cues: If the $k$ nearest objects of the focal agent are other agents, it will not be capable to detect a DS $l$ even if $d_{il} < 1$ (see Fig 1b). Note that in the case of vanishing density of DSs, $\rho_{DS} \to 0$ (homogeneous environment), the model reduces to a simple flocking model with so-called metric-free alignment interaction with k-nearest neighbors (see e.g. [20, 28, 36]) with the additional feature of informed individuals. If instead of topological, metric interactions are used, then the model reduces in the limit of $\rho_{DS} \to 0$ and $R_{inf} = 0$ to a Vicsek-type alignment model [3], which has been extensively discussed in [37, 38]. For $R_{inf} > 0$ it is closely related to the model explored in [6], while for $R_{inf} = 0$ and $\rho_{DS} > 0$ it reduces to the model studied in [33, 34].

## Interaction networks and collective accuracy

In order to quantify the emergent collective dynamics and study the effect of varying attention capability, we have performed systematically numerical simulations of the above model for varying attention limit $k$, DS density $\rho_{DS}$, and the ratio of informed individuals $R_{\inf}$ (see Methods for details). By neglecting the directed nature of inter-individual links, the entire agent system can be viewed as a time-dependent, undirected interaction network. For all DS densities, we obtain the same qualitative picture in the stationary state: For low $k$, we observe strongly fragmented dynamical networks, which at given time $t$ are characterized by a large number of small, disconnected sub-groups (see Fig 2a and S1 Fig). Each such cluster corresponds to an isolated connected component. These components are not static: We observe continuous fission-fusion of clusters over time due to randomness in individual motion and interactions with DSs (see S1 Video). By increasing the attention limit $k$, we observe a fast decrease in the number of disconnected clusters that results in an increase in average cluster size (see Fig 2b and S2a Fig). Eventually, by increasing $k$ above a critical value, we can obtain fully connected networks with a single connected component (see S1 Text, Sec. I and S2b Fig). Correspondingly, the average life time of a connection between specific agents grows strongly with increasing attention limit $k$, whereas an increase in DS density $\rho_{DS}$ reduces the life time of individual edges in the network (see S2c Fig). In general, as one would intuitively expect, the network of social interactions becomes more tightly connected with increasing attention capacity $k$.

We measure the accuracy of the collective migration through the average agreement between the heading $\hat{u}_j$ of individuals and the preferred direction of motion $\hat{u}_p$:

$$C = \left\langle \frac{1}{N} \sum_j \hat{u}_j \cdot \hat{u}_p \right\rangle \tag{1}$$

with $\langle \cdot \rangle$ indicating the temporal average in the stationary state. If all agents move perfectly along the preferred direction, which is available only to informed individuals, then we obtain $C = 1$, whereas for disordered movement we observe $C \approx 0$.

For collective behavior in homogeneous environments it has been shown that increasing the connectivity of the interaction networks is beneficial for coordination [29, 30], which is also in line with general results on synchronization in (dynamic) networks of oscillators [39, 40]. In particular, for few informed individuals, one intuitively expects that a strongly connected information network ensures that information about the preferred direction of motion diffuses more efficiently across large parts of the collective. Therefore, the natural prediction would be that collective accuracy increases with increasing attention capacity $k$. This is indeed the case in the limit of vanishing density of DSs, $\rho_{DS} = 0$ (homogeneous environment, see Fig 3a), where we observe a monotonous increase in accuracy $C$ with the attention capacity $k$, for a fixed ratio of informed individuals: Whereas for $k = 1$, in order to achieve an accuracy of $C > 0.9$, we require the majority of the collective to be informed ($R_{\text{inf}} \geq 0.6$), for $k = 6$ it is already sufficient to have a small fraction $R_{\text{inf}} \geq 0.2$ of informed individuals to achieve the same level of collective accuracy.

The situation completely reverses for high DS densities (see Fig 3b). For $\rho_{DS} = 0.2$ we observe a monotonous decrease in the collective accuracy with increasing $k$ for all values $R_{\text{inf}} > 0$. In order to achieve a certain level of collective accuracy (e.g. $C = 0.5$) for larger $k$ we need a larger fraction of the system to be informed. In other words, stronger connected flocks become more difficult to guide. Even more dramatically, for the largest attention capacity investigated ($k = 24$), the maximal attainable average accuracy for a fully informed system ($R_{\text{inf}} = 1$) is $C \approx 0.62$. This is lower than the average collective accuracy of $C \approx 0.66$ for collectives at minimal attention capacity ($k = 1$) with only a tiny fraction of informed individuals $R_{\text{inf}} \approx 0.013$ (1.3% of the entire collective). In fact, it appears that for $k = 1$, the collective accuracy $C$ versus $R_{\text{inf}}$ depends only very weakly on the DS density, whereas the accuracy for high $k$ is massively decreased for all $R_{\text{inf}} > 0$.

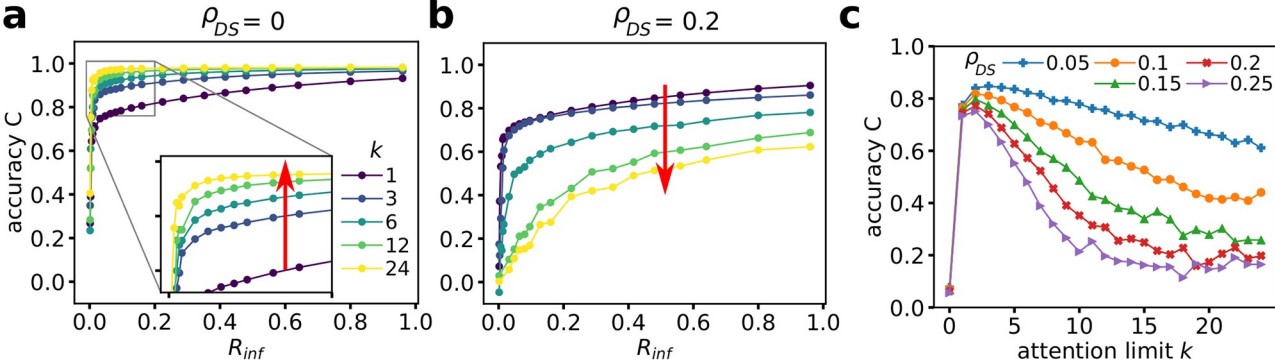

**Fig 3. Collective accuracy.** Collective accuracy $C$ of migration along the preferred direction versus the ratio of informed individuals for different attention limits $k$, for environments with no danger sites $\rho_{DS} = 0$ (**a**), and environments with high DS density $\rho_{DS} = 0.2$ (**b**). The red arrows show the direction of increasing $k$. **c**: Collective accuracy $C$ versus attention limit $k$ for different DS densities $\rho_{DS}$ at $R_{inf} = 0.1$.

Thus, in complex environments, strongly limited attention resulting in very sparse, fragmented—yet dynamic—interaction networks, turns out to be highly beneficial for global accuracy (see Fig 3c). The counter-intuitive effect of high accuracy for sparsely connected networks can be understood through an analysis of information flows and how environmental information is processed by the collective, for example through analysis of the (stationary) probability distribution of agents having a particular combination of in and out-degree. The out-degree $D_{out}$ quantifies how many individuals a focal individual pays attention to, whereas the in-degree $D_{in}$ is the number of others paying attention to the focal individual. Agents directly responding to a DS have a social out-degree $D_{out} = 0$—they ignore their neighbors. However, their neighbors can still pay attention to them so their social in-degree $D_{in}$ can be larger than zero (see S1 Text, Sec I and Fig 2c and 2d), in this case they "broadcast" information on the DS through their evasion behavior to others. Therefore, environmental cues affect collective behavior directly through the individuals directly responding to a DS, as well as, indirectly through information transmitted to other agents via social interactions. High accuracy in complex environments requires effective self-isolation of the collective from distracting environmental cues. It can emerge due to two mechanisms: 1) The number of direct responders remains small; 2) Their influence on others is weak due to a low $D_{in}$, in particular compared to the in-degree of non-responders. In our case both effects play a role: For low $k$, the fraction of agents directly responding to DSs remains low even at very high $\rho_{DS}$. At low $k$, aligned individuals move together in dense sub-groups, and even if one or more agents enter a repulsion zone, there is a high probability that there are $k$ neighbors closer than the DS, which prevents the detection of the latter. In addition, the in-degree of agents responding to DSs is on average significantly lower than that of agents not responding to the DSs (Fig 2c and 2d). Agents evading a DS, have a high probability to move away from their neighbors, which in turn decreases the probability that they will be within the $kNO$ of others. Thus, in particular for low $k$, the indirect response to DSs, mediated through social interactions, is strongly inhibited. The emergent small, dense agent clusters at low $k$ permanently merge and split up over time, which leads to exchange of directional information across the collective on long time scales, eventually leading to high migration accuracy and high coordination levels (see S1 Video).

This situation changes with increasing attention capacity $k$. The chance of an agent to detect a DS increases strongly, as the distance to the $k$ nearest objects is an increasing function of $k$. As soon as this distance becomes larger than the distance to the next DS ($d_{il} \lesssim 1$), agents are capable to detect the environmental cues reliably, and react to them if they are within their repulsion zone. In addition, larger $k$ also increases the number of other agents paying attention to a direct-responder. The motion of the emergent sub-groups is still well coordinated at a local scale (see S2 Video). However, these sub-groups interact now predominantly with the environment by changing their direction of motion and by complex fission-fusion dynamics directly triggered by the DSs. This results in quick loss of directional information across time and space, and in a vanishing impact of the directional information provided by the informed individuals, which yields a strong decrease in collective accuracy $C$.

The contribution of both effects to the emergent self-isolation for low $k$ is shown in Fig 4a: The fraction of direct responders $r_d$ is much lower for low $k$ and shows only a slow increase with increasing DS density $\rho_{DS}$. The same holds for the fraction of first-order indirect responders $r_i$, defined by agents paying attention to at least one direct responder. For $k = 2$, $r_i < 0.1$ for all DS densities studied, whereas for $k = 24$ at high DS densities, it saturates at more than half of the entire collective ($r_i \approx 0.55$).

We note that in random environments the consensus direction always coincides with the preferred direction of informed individuals, i.e. high accuracy implies strong (directional) consensus and vice versa.

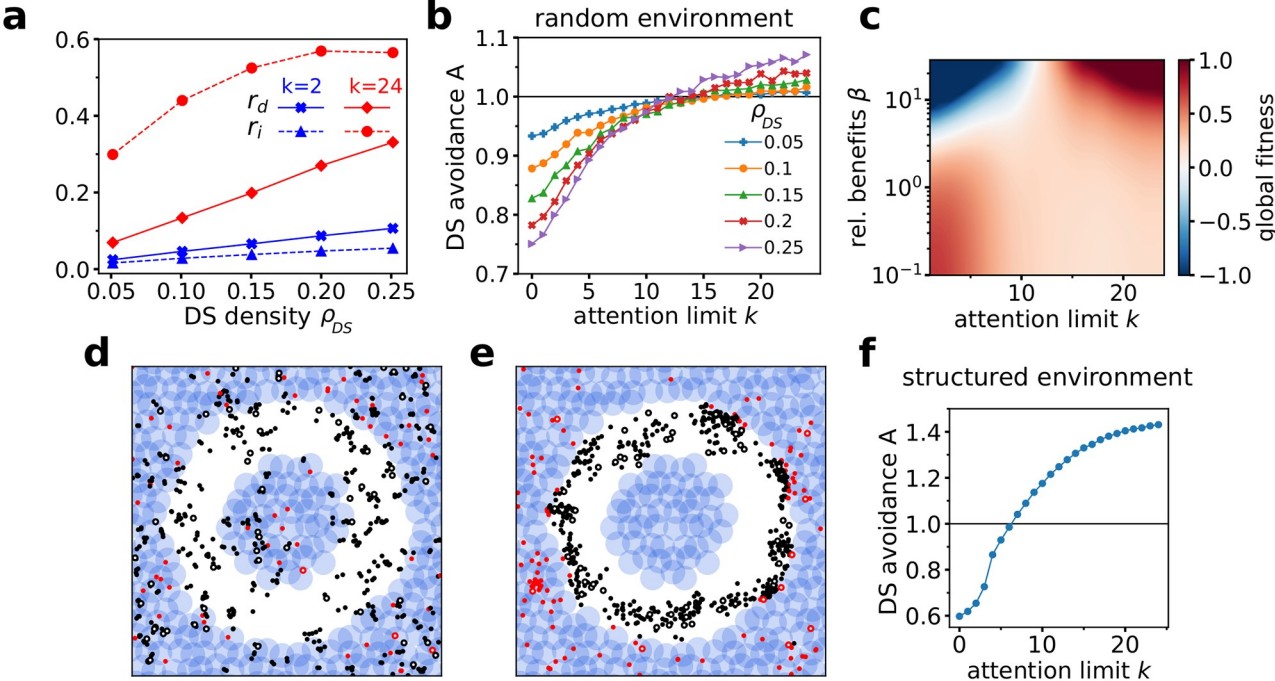

**Fig 4. Collective response to environmental cues. a**: The fraction of agents responding to DS directly $r_d$ (direct responders, solid lines), or indirectly via social interactions with direct responders $r_i$ (indirect responders, dashed lines), for $k = 2$ (blue) and $k = 24$ (red) versus DS density $\rho_{DS}$. **b**: DS avoidance $A$ versus attention limit $k$ for different DS densities $\rho_{DS}$. $A = 1$ corresponds to the DS avoidance of solitary (non-interacting) agents. **c**: Global fitness versus attention limit $k$ and relative benefits of DS avoidance $\beta$ at $\rho_{DS} = 0.25$. Red (blue) regions correspond to better (worse) performance of a collective than isolated individuals according to the fitness function used. **d** and **e**: Example snapshots of emergent collective behavior in structured environments with a circular, DS-free path. For low attention capacity ($k = 1$, **e**), individuals ignore the structure of the environment and align with the preferred direction of migration. At high attention capacity ($k = 16$, **f**), the collective behavior is dominated by the environmental structure and collective migration breaks down. **f**: DS avoidance $A$ for the structured environment depicted in **d**, **e** versus attention limit $k$. $A = 1$ is the DS avoidance of solitary agents in the same environment. For all panels: $R_{inf} = 0.1$.

In order to confirm that the observed dependence of accuracy on $k$ is a truly collective effect, which cannot be trivially traced back to the behavior of informed agents alone, we compared our results for accuracy of socially interacting agents with the case where the social interaction strength is set to zero: $\gamma_s = 0$ (see S1 Text, Sec II). For all $k$ we observe a massive increase in accuracy (see S3a Fig), showing that social interactions are crucial. We also analyzed the accuracy of uninformed and informed agents separately (see S3b and S3c Fig). Interestingly, the accuracy of informed individuals is increased for all $k$ values in a socially interacting system, whereby most interactions are with uninformed individuals ($R_{inf} = 0.1$). Uninformed individuals act as a directional "information reservoir": Once an informed individual becomes distracted due to direct interaction with the local environment, social interactions with other agents help it to quickly align back with the consensus direction, which coincides with the preferred direction of motion.

## Collective avoidance of repulsion zones

While limited attention is beneficial for consensus formation in complex environments, it may be very detrimental to the collective if the environmental cues (DSs) provide reliable information about environmental dangers (e.g. predators).

In order to quantify the performance of the collective in responding to environmental cues, we introduce in the following a DS avoidance measure. First, we compute the average fraction

of agents in "safe" areas, i.e. outside the DS repulsion zone. As a reference, let us estimate the same quantity in a control numerical experiment with the same number of agents but who are only interacting exclusively with the environment and not with other agents. Our DS avoidance measure $A$ is then defined as the ratio between these two quantities (see Methods for details). This measure $A$ allows us not only to compare the relative collective performance of the flock at different attention capacities $k$, but also to compare it with the performance of solitary agents without any social interactions. For $A > 1$, social interactions provide a benefit with respect to solitary individuals. For $A < 1$, a collective performs on average worse than solitary agents in avoiding the potentially dangerous areas. Fig 4b shows clearly that for small values of $k$, the collective performs much worse than solitary individuals in avoiding the repulsion zones. Here, increasing $k$ results in a monotonous increase in $A$. However, we observe $A > 1$ only for sufficiently large $k \gtrsim 12$. For lower $k$ it turns out that social interactions are detrimental with respect to DS avoidance. The observed behavior is directly linked to the emergent local echo-chamber effect at small $k$. Socially interacting individuals can only outperform solitary agents in responding to environmental cues, if sufficient amount of information is able to enter the interaction network and spread effectively through it.

We can quantify the emergent trade-off between migration accuracy (quantified by $C$) and DS avoidance (measured by $A$) through a global fitness function $F(C, A)$ (see Methods and S1 Text, Sec III for details). While $F$ depends implicitly on the attention limit $k$ through $A$ and $C$, we introduce the parameter $\beta$ to quantify the relative benefits of avoidance versus accuracy. Whereas in safe environments, benefit of avoidance may be negligible ($\beta \approx 0$), in environments where local cues provide important information about potential dangers one can assume $\beta \gg 1$. Fig 4c shows $F$ as a function of $k$ and $\beta$ for collectives in random DS fields. We note that $F = 0$ corresponds to the average fitness expected for solitary individuals. For low $\beta$, we observe a single maximum of $F$ at low attention capacities $k \approx 2$. For increasing benefits of DS avoidance $\beta$, a second maximum emerges at large $k$, while at low $k$, the socially interacting collectives are on average outperformed by solitary individuals.

This trade-off between accuracy and avoidance, or "responsiveness", becomes particularly prominent if we consider structured, inhomogeneous environments containing free paths and voids in a landscape otherwise filled with high density of DSs, instead of random environments. At low $k$ the agents completely ignore the environmental structure and their dynamics are dominated by social interactions with high directional accuracy. At high $k$, the situation reverses and the collective dynamics is dominated by the environment, where agents track the environmental structure, staying preferentially in areas with low DS density, while ignoring the preferred direction of migration (see Fig 4d–4f, see also S4 Fig, S3 and S4 Videos).

## Model variations and generality of results

So far, we have considered social interactions in which agents do not pay special attention to neighbors responding to DSs. This is motivated by the idea of social interactions being based on observations of behavior of others but absence of direct communication about the cause of their particular behavior. In order to test the robustness of our results, we explored an extension of the basic model, by introducing active signaling about potential danger by agents interacting with a DS. In this case, all neighboring agents connected to the signaler put their full attention on the signaling agent and respond only to it, while ignoring other non-signaling agents. As expected, the additional signaling improves the collective DS avoidance due to increased saliency of the corresponding cues within the network. However, the general results—in particular the coordination and responsiveness trade-off—remain unchanged (see S5 Fig).

An important assumption of our model is that individuals treat social and environmental cues exactly in the same way: Other agents and DSs must be within the kNO in order to be detected. However, if DSs signal potential danger it can be argued that individuals should be more sensitive to corresponding environmental cues. Thus their attention should be biased towards DS detection, i.e. they should have a higher probability to detect and respond to a DS, once they are within the corresponding avoidance zone. We confirmed that our general results hold for such a bias in individual attention through a minimal model extension: We assume that at any given time, an individual within an avoidance zone can directly detect the corresponding DS with probability $P_{direct}$, independent of its social neighborhood, i.e. irrespective whether the DS is among its k-nearest objects. For $P_{direct} = 0$ we recover our original model, whereas $P_{direct} = 1$ corresponds to perfect detection of DS, effectively switching off any attentional restrictions on interactions with the local environmental cues. Our results show that as long as there is some attentional "interference" between social and environmental cues $P_{direct} < 1$, our qualitative results on the coordination-responsiveness trade-off remain unchanged (see S1 Text, Sec IV and S6 Fig for more details).

Furthermore, the general trade-off between consensus and responsiveness to DSs does not depend on the presence of informed individuals. For $R_{\mathrm{inf}} = 0$ and $\rho_{DS} = 0$, our system reduces to a Vicsek-type model with topological interactions in homogeneous environments [27, 28, 36]. With $R_{\mathrm{inf}} = 0$ and $\rho_{DS} > 0$, i.e. without the bias provided by informed individuals, the model becomes a topological Vicsek-like model in heterogeneous environments. All the phenomena discussed above hold also in this case, if we replace the collective accuracy $C$ by the (normalized) average velocity $\tilde{C} = (\sum_i \hat{u}_i)/N$, which quantifies the overall degree of consensus, i.e. (orientational) order, in the system (see S1 Text, Sec V and S7 Fig). It is worth stressing that there exist fundamental differences with metric Vicsek-like models in heterogeneous environments (cf. [33, 34]), where agents are not subject to any cognitive limitation and display exclusively a limited perception capacity. In addition, variations of the topological interaction mechanism do not affect the reported results. Specifically, we confirmed that we obtain the same collective effects in a model where the $k$-nearest objects are selected from the Voronoi neighborhoods (S8 Fig). This version of the model resembles the spatially balanced topological flocking algorithm proposed in [41] and represents a better approximation of visual networks [14]. The robustness of our results with respect to the exact topological model shows that our results are not affected by rare configurations of the kNN-model, which may be incompatible with visual interactions. A more detailed discussion is given in S1 Text, Sec VI.

All this suggests that the fundamental coordination-responsiveness trade-off discussed here is independent of the specific choice of the social interaction model.

## Discussion

Using a generic flocking model we have demonstrated the importance of finite attention capacity of individuals for collective information processing in complex environments. In our model, agents dynamically allocate their limited attention to process social and environmental stimuli, whereby the saliency of different stimuli is governed by their spatial vicinity. We demonstrated that contrary to the general intuition, large-scale coordination and, as a result, the accuracy of collective migration in complex environments is maximized for strongly limited individual attention capacity. High levels of accuracy for agents which can pay attention only to few stimuli at a time, are a direct consequence of a self-isolation from distracting environmental cues through social interactions. This comes at a price of a strongly inhibited response of the collective to distraction sites, as quantified by the avoidance parameter. On the other hand, the increased ability of agents to respond collectively to environmental cues for high

attentional capacity leads to a breakdown of collective accuracy. In this case the strong information inflow through local distractions overrides the information on the global preferred direction of motion available only to a minority of informed individuals. This higher sensitivity to the environment results in a better performance of collectives versus solitary agents in terms of the avoidance parameter. This demonstrates a fundamental trade-off between large-scale coordination and collective accuracy on the one hand, and the dynamical response to local environmental cues in complex environments, on the other hand. We would like to emphasize that our general finding of weaker connected networks achieving higher global accuracy in complex environments is diametrically opposed to widely accepted and intuitive knowledge in network science that more connections lead more effective information exchange and thus higher levels of synchronization (see e.g. [29, 30, 39, 40]). We recover this intuitive result for flocking in empty environments, which demonstrates how taking into account environmental disturbances may dramatically change the collective behavior of self-organizing systems.

Our results suggest a specific link between cognitive and sensory capabilities of flocking animals and the ecological context. For example, for migrating animals, with high fitness benefits associated with coordination and information sharing on a preferred migration direction, with no (or very low) fitness costs of ignoring local environmental cues, strongly limited attention appears to be beneficial. However, if collective response to environmental cues is highly relevant for individual fitness but global coordination is not, as for example in foraging reef fish [42], then being able to pay attention to many stimuli simultaneously becomes important. This yields testable hypotheses, on how the attention capability of different species exhibiting grouping behavior should co-vary with ecological niche, or how individuals within the same species should modulate their attention capabilities across contexts. Here, we note that a recent analysis of collective behavior in *Hemigrammus rhodostomus*, a strongly schooling fish species, suggests that an individual fish appears to pay attention only to one or two neighbors at a time [21].

Interestingly, being social offers an advantage over solitary behavior with respect to response to DSs only above a critical attention capacity. This poses some fundamental questions regarding the co-evolution of social behavior and individual attention capacity, especially taking into account potential developmental costs of higher attention capacity. Overall, our results point towards a complex interrelation between pre-existing attention capability, evolution of grouping behavior, and the ecological niche.

We note that in our minimal model we varied only a single dimension of cognitive capabilities, namely the total number of objects an individual can pay attention to. There are other more complex cognitive processes, which affect the individual processing of a large number of social and non-social stimuli. For example, object recognition and classification, may enable individuals to dynamically vary the relative saliency of social and non-social stimuli, which is not considered in this work but could allow individuals to adapt to different behavioral contexts. In general, the strength of the observed trade-off will depend on model choice and model details. For example, making the agents more likely to detect danger sites, will make the collectives more responsive to the environmental cues. However, if the overall attention capacity has an upper bound—as assumed here—it must come at the cost of decreasing social interactions. Hence, the existence of the general effects discussed here, in particular the surprising increase in collective accuracy with decreasing attention capabilities, is not restricted to the particular model choice. The qualitative results should hold as long as the following three conditions are met: 1) the attention capacity is limited, 2) the salience of social cues has some distance-based component, and 3) the structure of the interaction network emerges naturally from spatial self-organization. Overall, this work demonstrates the fundamental importance of

potential constraints in sensory and cognitive abilities of individuals on emergence and function of collective behavior.

We considered explicitly the case of spatial flocking behavior, however one can speculate that the coordination-responsiveness trade-off represents a more general principle, and should be observable in different collective behaviors. Only recently it was shown that Guinea baboon exhibit stronger response to known social cues than to novel ones. It was hypothesized that this unexpected behavior can be linked to the complex social environments in which Guinea baboon groups live, and the corresponding necessity to filter out irrelevant or distracting information ("social noise") [43]. Our findings have also potential implications for the design of interaction networks in artificial distributed systems, such as robotic swarms that operate in complex environments. Instead of continuously increasing the sensory and computational abilities of individual agents in order to cope with the consensus problems in complex environments, it may be promising to think about constraining the "cognition" of swarming robots. By generating specifically tailored attentional bottlenecks, resulting in emergent self-isolation as observed here, one can facilitate coordination and exchange of relevant information in complex environments. Attentional bottlenecks based on static features, e.g. colors, which can be easily distinguished from the background, are widely used in swarm robotics [44]. Here, we demonstrated that dynamical features (as opposed to static ones), like relative distance, or relative speed [18], could provide effective means of coordination in complex environments, where "filtering" based on static features is difficult or not feasible.

Last but not least, our work yields potentially interesting implication for social sciences, where "echo-chambers" have received considerable attention recently. In human social networks, this effect is typically linked to *homophily* and *confirmation bias* [45, 46]. Our results show collective self-isolation from conflicting external information as agents moving in the same direction self-organize into tightly interacting social groups. This can be viewed as an "echo-chamber"-like effect, which emerges naturally even in the absence of such explicit biases and self-sorting mechanisms as homophily. On the one hand, this suggests that these self-isolation tendencies may be much more prevalent and easier to obtain for agents with limited attention capacity. On the other hand, our results provide support for the evolution of proximate, socio-psychological mechanism facilitating the formation of echo chambers, such as homophily, by demonstrating how an emergent "echo chamber"-like effect strongly increases intra-group synchronization for a collective in a complex environment.

## Methods

### Agent-based model

We consider a system of $N$ self-propelled agents and $N_{DS}$ danger sites DSs in a two dimensional domain of size $L \times L$ with periodic boundary conditions (torus). The agent and DS densities are thus $\rho = N/L^2$ and $\rho_{DS} = N_{DS}/L^2$. The agents move with a fixed speed $v_0 = 0.5$ and respond to other agents and DSs by changing their direction of motion $\hat{u}_i = (\cos \varphi_i(t), \sin \varphi_i(t))^T$. The behavior of each agent is mathematically described by the following stochastic equations of motion (see Section VII in S1 Text), whereby $d\varphi_i/dt$ corresponds to the turning rate of agent $i$:

$$\frac{d\vec{x}_i}{dt} = \vec{v}_i(t) = v_0 \hat{u}_i(t) = v_0 \begin{pmatrix} \cos \varphi_i(t) \\ \sin \varphi_i(t) \end{pmatrix} \tag{2}$$

$$\frac{d\varphi_i}{dt} = (1 - g_i(t)) \left[ \frac{\gamma_s}{n_s} \sum_{j \in kNO} \sin(\varphi_j - \varphi_i) - \gamma_p sin(\varphi_i) \right]$$

$$+ g_i(t) \frac{\gamma_l}{n_l} \sum_{l \in kNO} \sin(\alpha_{i,l} - \varphi_i) + \eta \xi_i(t) \qquad (3)$$

The first term in the turning response Eq 3, is the alignment interaction. A focal individual aligns with the strength $\gamma_s = 1$ with neighbors $j$, which are part of its kNO set. In addition, informed individuals have a (weak) tendency $\gamma_p = 0.1$ to move in a preferred direction, here $+\hat{x} = (1, 0)^T$ ($\gamma_p = 0$ for non-informed individuals). The turning away (repulsion) from the DSs $l$, which are in the kNO set is given by the third term ($\gamma_l = 1$). Here, $\alpha_{i,l}$ is the spatial position angle of the focal agent relative to the DS $l$. Both interactions are normalized by the number of agents and DSs, respectively ($n_y = \sum_{y \in kNO} 1$). The function $g_i(t)$ determines whether the agent responds to DSs, or whether it aligns with its neighbors, and, in the case of informed individuals, biases its motion towards the preferred direction of motion. It is defined as

$$g_i(t) = \begin{cases} 1 & \text{for} \quad l \in kNO \quad \text{and} \quad d_{il} < r \\ 0 & \text{else.} \end{cases} \qquad (4)$$

Note that an agent responding to a non-social cue ($g(t) = 1$), will not interact socially with other agents until its interaction with the DS is terminated, either because it leaves the repulsion zone or the site falls out of its $k$-nearest object set. The last term in Eq 3 accounts for the stochasticity in the motion of individuals, with $\eta$ being the angular noise strength and $\xi_i(t)$ a normally distributed Gaussian white noise with standard deviation 1. In all the simulations discussed, $\eta = 0.25$ and the average density of agents is fixed at 1 in a box of linear size $L = 25$ ($N = 625$). If not stated otherwise, the fraction of informed individuals is $R_{inf} = 0.1$. We confirm that changing the model parameters, $v_0$, $\eta$, $L$, $\gamma_{(p,l,s)}$, and $\rho$ does not change our general qualitative results, i.e. the presence of a maximum accuracy for low $k$ values and increase in DS avoidance with $k$. The results for some of these variations are represented in S9 Fig. See S1 Text, Sec VII for additional details on numerical implementation.

## Avoidance parameter

We quantify avoidance of repulsion zones through $\tilde{A} = 1 - \langle N_{rz}(t)/N \rangle$. Here $N_{rz}(t)$ is the number of agents, which are within at least one repulsion zone at time $t$, and $\langle \cdot \rangle$ represents temporal average in the stationary state. $\tilde{A}$ will always decrease with DS density, as more and more space is occupied by repulsion zones. In order to control for this trivial effect, we rescale $\tilde{A}$ by the corresponding value for non-interacting agents $A = \tilde{A}/\tilde{A}_{ni}$. Thus, $A = 1$ indicates same average performance of the flock as solitary agents without any social interactions. Here, solitary individuals are always responding to a DS, once they are within the corresponding repulsion zone.

## Fitness function

We can quantify the emergent trade-off between collective accuracy and responsiveness, through the following global fitness function depending on the collective accuracy C and DS avoidance:

$$F(C, A) = C + \beta(A - 1), \qquad (5)$$

with $\beta$ being the relative benefits of DS avoidance with respect to migration accuracy. The above function was defined in a way so that a value $F = 0$ corresponds to the behavior of solitary individuals, where the average accuracy vanishes ($C = 0$) and the DS avoidance is $A = 1$, according to the definition above. Thus, only for $F > 0$ the collectives perform better than single individuals (see S1 Text, Sec III for more details).

## Supporting information

**S1 Text.** Sections: I. Dynamical, directed interaction networks. II. Coordination-responsiveness trade-off as an emergent collective effect. III. Quantifying the coordination-responsiveness trade-off—global fitness function. IV. Modified model with higher priority of DS avoidance. V. Emergence of global order in the absence of informed individuals. VI. Discussion of idealizing model assumptions in relation to visual interactions. VII. Numerical implementation and experiments.
(PDF)

**S1 Video. Flocks with low attention capacity in a random environment.** Collective behavior at high density of DSs for low attention capacity $k = 1$ characterized by high accuracy of collective migration.
(MP4)

**S2 Video. Flocks with high attention capacity in a random environment.** Collective behavior at high DS densities for high attention capacity $k = 24$ characterized by efficient response to environmental cues.
(MP4)

**S3 Video. Flocks with low attention capacity in a structured environment.** Collective behavior in structured environment with a circular DS-free region for low attention capacity $k = 1$.
(MP4)

**S4 Video. Flocks with high attention capacity in a structured environment.** Collective behavior in structured environment with a circular DS-free region for high attention capacity $k = 24$.
(MP4)

**S1 Fig. Full interaction networks.** Snapshots of the (undirected) social interaction network in random environments with $\rho_{DS} = 0.25$ for $k = 3$ upper panel, and $k = 12$ lower panel, at $R_{inf} = 0.1$. Black agents are socially interacting, and red agents react to DSs. Informed and uninformed individuals are represented by empty and filled circles, respectively. Light blue circles are repulsion zones of DSs specified with blue dots. For the sake of clarity, the links between agents interacting with their periodic neighbors are removed. The black squares depict the close-ups shown in panels a, b of Fig 2 (main text). For low attention capacity ($k = 3$) the network is sparse, composed of many small connected components, whereas for large attention capacity ($k = 12$), the network is highly connected with less components.
(PDF)

**S2 Fig. Temporal interaction networks. a**: Average number of connected components (CC) of the interaction network versus attention limit $k$. For all DS densities $\rho_{DS}$, we observe a fast decay of the number of connected components, which due to constant number of agents $N$ is equivalent to the growth of the average connected component size, indicating a more tightly connected temporal network. **b**: The probability of observing one connected component during simulation. By increasing DS density, nonzero probability happens at larger k values. **c**:

The average life time of an edge in the interaction network decreases with increasing density of DSs. However with increasing $k$ for a fixed DS density, we observe longer life times due to increased connectivity in the interaction network.
(PDF)

**S3 Fig. Emergent collective behavior.** Accuracy C vs attention limit $k$ calculated for the whole system (**a**), uninformed individuals (**b**) and informed individuals (**c**). Solid lines are for the interacting system and dashed dotted lines are for the non-interacting system with $\gamma_s = 0$.
(PDF)

**S4 Fig. Collective accuracy and DS avoidance in a structured environment with circular DS-free region.** Accuracy $C$ (triangles) and DS avoidance $A$ (circles) versus attention limit $k$ (**a**). The horizontal line, $A = 1$, corresponds to DS avoidance of non-interacting agents. For socially interacting agents with low $k$ values ($k = 1, 2$), we observe high accuracy $C$ together with almost complete ignorance towards environmental cues (see S3 Video). By increasing $k$, more agents start to sense the environment and react to DSs. At high $k$, the collective behavior is fully determined by the local environmental features: We observe collective rotation along the circular path and complete ignorance of the global migration direction accessible to informed individuals (see S4 Video). This trade-off is shown quantitatively by the global fitness function in panel **b** versus attention capacity $k$ and relative DS avoidance benefit $\beta$. There are two maxima in global fitness, one for low $k$, $\beta \ll 1$, showing migration accuracy to be beneficial for the group, the other at high $k$, and $\beta > 1$, which indicates higher benefits associated with DS avoidance in comparison to collective accuracy.
(PDF)

**S5 Fig. Collective coordination-responsiveness trade-off with active signaling.** Each agent connected to another individual signaling direct interaction with a DS (direct responder), pays only attention to the signaler(s) and ignores other social cues. Accuracy $C$ (**a**) and DS avoidance $A$ (**b**) versus attention limit $k$ for different DS densities at $R_{inf} = 0.1$.
(PDF)

**S6 Fig. Model extension with the priority of DS avoidance.** Accuracy C and DS avoidance A vs attention limit $k$ in a model where the agents first detect DSs with some probability and otherwise interact with their kNO. **a**: $P_{direct} = 0.2$, **b**: $P_{direct} = 0.5$.
(PDF)

**S7 Fig. Emergence of global order in the system with no informed individuals, $R_{inf} = 0$. a**: Coordination $\tilde{C}$ (directional order) versus attention limit $k$ for different DS densities. **b**: DS avoidance versus attention limit $k$. $A = 1$ corresponds to non-interacting agents. The qualitative behavior with a coordination-responsiveness trade-off is similar to the model with informed individuals, but here instead of a specific direction, the emergent consensus direction is random (spontaneous symmetry breaking).
(PDF)

**S8 Fig. Collective motion of agents with Voronoi-based kNN interaction network.** For each focal agent $k$ nearest neighbors are selected from first shell of Voronoi neighbors. If the number of neighbors in first layer is smaller than $k$, then depending on $k$, the second Voronoi shell is considered. It is defined by the Voronoi neighborhood of the (direct) Voronoi neighbors of the focal agent. Accuracy $C$ (**a**) and DS avoidance $A$ (**b**) versus attention limit $k$ at $R_{inf} = 0.1$.
(PDF)

**S9 Fig. Robustness of the general results for *C* and *A* vs *k* with respect to variation of model parameters.** The panels show results for changing individual model parameters by a factor of 2 (left columns) and 0.5 (right columns). Different rows represent variation of different parameters, from top to bottom: $v_0, \gamma_s, \gamma_l, \gamma_p$. The non-varied parameters are always set to the default parameters: $v_0 = 0.5, \gamma_s = 1, \gamma_l = 1, \gamma_p = 0.1$.
(PDF)

## Acknowledgments

We thank Ana Sofía Peruani for providing the artwork in Fig 1a.

## Author Contributions

**Conceptualization:** Parisa Rahmani, Fernando Peruani, Pawel Romanczuk.

**Data curation:** Parisa Rahmani.

**Formal analysis:** Parisa Rahmani, Fernando Peruani, Pawel Romanczuk.

**Funding acquisition:** Pawel Romanczuk.

**Investigation:** Parisa Rahmani, Fernando Peruani, Pawel Romanczuk.

**Methodology:** Fernando Peruani, Pawel Romanczuk.

**Project administration:** Pawel Romanczuk.

**Software:** Parisa Rahmani, Pawel Romanczuk.

**Supervision:** Fernando Peruani, Pawel Romanczuk.

**Visualization:** Parisa Rahmani, Pawel Romanczuk.

**Writing – original draft:** Pawel Romanczuk.

**Writing – review & editing:** Parisa Rahmani, Fernando Peruani, Pawel Romanczuk.

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
