## [Decision Letter · Decision Letter 0]

28 Oct 2019

Dear Dr Romanczuk,

Thank you very much for submitting your manuscript 'Flocking in Complex Environments – Attention Trade-offs in Collective Information Processing' for review by PLOS Computational Biology. Your manuscript has been fully evaluated by the PLOS Computational Biology editorial team and in this case also by independent peer reviewers. The reviewers appreciated the attention to an important problem, but raised some substantial concerns about the manuscript as it currently stands. While your manuscript cannot be accepted in its present form, we are willing to consider a revised version in which the issues raised by the reviewers have been adequately addressed. We cannot, of course, promise publication at that time.

Sincerely,

Jessica C. Flack

Associate Editor

PLOS Computational Biology

Stefano Allesina

Deputy Editor

PLOS Computational Biology

[LINK]

Both reviewers feel this is an interesting manuscript with a lot of potential. However both also have concerns that need to be addressed in a revision. In particular please pay attention to the comments of reviewer 2 regarding the empirical justification for the model and the issue of what constitutes a genuine collective phenomenon. When you submit your revision please include a point by point response to both reviewers' comments. Thank you.

Reviewer's Responses to Questions

**Comments to the Authors:**

Reviewer #1: The authors studied a Vicsek like model of flocking in environments with obstacles. The novelty of the model is placing a cap on the number of interacting neighbors (k). The model is studied systematically under different density of avoidance zones and values of k, and reveals a fairly unintuitive result: for a wide range of obstacle densities, systems with low k's achieve higher accuracy than systems with larger k values.

Overall I find this to be an excellent paper: the topic is of wide interest, the text is well written, and the methods are sound. However, I do have a few comments that I hope the authors could address:

(1) This is a minor one. The terms "heterogeneous environments" and "homogenous environments" are frequently used in the text, but they are not well defined. It would be worth while mentioning in the introduction that you will be referring to simulation with obstacles (or distraction sites) as "heterogeneous environments".

(2) The authors define the accuracy of the collective migration as the average deviation from the preferred direction of motion u_p. But they do not take into account the avoidance of obstacles in their accuracy measure. Wouldn't avoiding those "danger zones" be beneficial and even crucial for the success of the flock? I suggest to include a discussion of the success of avoiding the obstacles.

(3) By setting the maximal number of neighbors an individual can pay attention to, the simulations are prone to large fluctuations in the lists of interacting neighbors. Depending on the configuration, an individual could switch attention from one neighbor to another, even if those neighbors are separated by a large spatial distance. This should be mentioned somewhere in the text, and preferably quantified somewhere in the SI.

(4) When using high k values (say 24) and high obstacle densities (say 0.25), it's not clear that it is possible to obtain a line of site with all the interacting neighbors. The authors could check this in their simulations, and limit reporting results only to more physically realistic regimes.

Reviewer #2: Rahmani et al. introduce a model of collective motion with limited agent perception in spaces with danger areas. This model builds on well studied examples from the literature, in particular the flocking models of Vicsek and colleagues, using a topological interaction regime limited to k nearest-neighbours or danger sites. They also incorporate ‘informed individuals’ who have a preferred direction of motion. They study the collective motion of agents in this model as a function of the attention capacity (k) as well as the density of danger sites and the proportion of informed individuals. The key result is that the collective motion displays somewhat counter intuitive properties as the attention capacity is increased: group cohesion and accuracy breaks down as agents are driven primarily to avoid danger sites.

I found the main result of this paper interesting and potentially important, but I have concerns about its significance from both a biological and an analytical perspective.

Analytically, I think the authors could do more to demonstrate the observed effects are genuinely an emergent collective phenomenon. In particular, if we were to consider a model with a zero-strength social interaction, we would expect to see that ‘collective accuracy’ would be decreased ask increases as more and more informed individuals are disrupted, while uninformed individuals would experience no change from their random orientations. To what extent do these results go beyond this expectation? There is a partial answer to this in Figure 4 A, but I would have found it useful to see a breakdown in the effect on informed and uninformed individuals separately.

I was also unable to locate a justification for the assigned values of many model parameters, or a detailed discussion of how sensitive the model may be to these. For instance, L419 notes that the results are qualitatively unchanged under variation of nu_0, eta and L, but does not mention variation in the gamma parameters or in the density of agents (N/L^2)

Biologically my concern focuses on how realistically the model represents real systems of interest. Obviously the model is a theoretical abstraction, and absolute realism is neither expected nor desirable. However, one feature of the model struck me as possibly unrealistic: agents treat danger sites on an equal footing with other agents when allocating attention (i.e. a danger site must be within the k nearest objects to be attended to, even if one is within the danger radius), yet once the agent does attend to a danger site this overrides any social response. This seems inconsistent: if danger sites are overwhelmingly important to attend to, surely the agents ought to be more active in attending to their possible existence. The only explanation I could conceive of to explain the equal treatment is if interactions are visual, and attention is moderated by visual occlusion (and agents make no significant effort to overcome occlusion). I think the authors should discuss this assumption explicitly in the paper and potentially investigate the robustness of their results to alternative ways of allocating the limited attention. As it stands and based on this model alone, I can’t agree with the authors’ assertion that this ‘demonstrates a fundamental trade-off’ (L313-314).

In the discussion the authors point to several empirical studies in support of their central finding. I found the connection with ref. 43 (Guinea baboons) to be unconvincing as a comparison here. The authors point to ref 21 (Jiang et al) as an empirical example of fish attending to a limited set of neighbours, however the abstract of this paper states ‘we find no correlation between the distance rank of a neighbor and its likelihood to be influential’, which does not align with the model used here. I am also skeptical about the speculation regarding robot swarms. There seems to me no reason why perception itself need be limited in agents to produce cohesion and other desirable effects: agents could be programmed to respond to fewer stimuli than they can perceive, or to intelligently to different sources of information. I think the authors are on stronger ground when they state that their results generate predictions and testable hypotheses, and I would suggest that more focus is given to laying these out in some detail and discussing how they could be tested.

Minor points:

1. Would it better to define the density of danger sites as the actual proportion of space taken up by danger sites rather than the number of danger sites divided by L^2? For one thing, this would make results more comparable across different values of L.

2. Panels C and D of Figure 2 do not seem to be discussed anywhere in the main text.

3. The authors repeatedly refer to the effect of attention on collective ‘consensus’ (e.g L176), but the results pertain to collective ‘accuracy’. These are not the same; if all agents move coherently perpendicular to the preferred direction they would have consensus but not accuracy.

4. The use of delta to parameterise the relative benefits of danger site avoidance in equation 5 was confusing to me, as it looked like a dirac delta function. I would suggest using something less ambiguous.

5. L137 - what is the ‘critical value’ to create a connected graph? Is it consistent across simulations, and can it be compared to the typical value needed to connect uniformly scattered points (c. 3-4 in 2D I think)?

6. L263, should be dynamics ARE not dynamics IS

7. In equation 5 the costs/benefits associated with accuracy and avoidance and assumed to be linear. Can this be justified biologically, or is this just a convenience?

**Have all data underlying the figures and results presented in the manuscript been provided?**

Reviewer #1: Yes

Reviewer #2: Yes

PLOS authors have the option to publish the peer review history of their article (what does this mean?). If published, this will include your full peer review and any attached files.

Reviewer #1: No

Reviewer #2: No

---

## [Decision Letter · Decision Letter 1]

29 Jan 2020

Dear Pawel

We are pleased to inform you that your manuscript 'Flocking in Complex Environments – Attention Trade-offs in Collective Information Processing' has been provisionally accepted for publication in PLOS Computational Biology.

Before your manuscript can be formally accepted you will need to complete some formatting changes, which you will receive in a follow up email. A member of our team will be in touch within two working days with a set of requests.

Best regards,

Jessica C. Flack

Associate Editor

PLOS Computational Biology

Stefano Allesina

Deputy Editor

PLOS Computational Biology

Reviewer's Responses to Questions

**Comments to the Authors:**

Reviewer #1: The authors have done a thorough job on the revisions, matching the great job on the initial submission. I have no further comments.

Reviewer #2: The authors have satisfactorily addressed all of my concerns. Textual changes to the manuscript are welcome clarifications, and the additional supplementary materials make the case for a fundamental trade-off between perception and cohesion much stronger.

**Have all data underlying the figures and results presented in the manuscript been provided?**

Reviewer #1: Yes

Reviewer #2: Yes

PLOS authors have the option to publish the peer review history of their article (what does this mean?). If published, this will include your full peer review and any attached files.

Reviewer #1: No

Reviewer #2: No

---

## [Editor Report · Acceptance letter]

19 Mar 2020

PCOMPBIOL-D-19-01264R1 

Flocking in Complex Environments – Attention Trade-offs in Collective Information Processing

Dear Dr Romanczuk,

I am pleased to inform you that your manuscript has been formally accepted for publication in PLOS Computational Biology. Your manuscript is now with our production department and you will be notified of the publication date in due course.

With kind regards,

Laura Mallard
